# Trauma Functioning and Well-Being in Children Who Receive Mental Health Aid after Natural Disaster or War

**DOI:** 10.3390/children9070951

**Published:** 2022-06-25

**Authors:** Emily A. Simonds, Katrina Arlene P. Gobenciong, Jonathan E. Wilson, Michael R. Jiroutek, Nicole R. Nugent, Miranda A. L. van Tilburg

**Affiliations:** 1Department of Pharmaceutical and Clinical Sciences, College of Pharmacy & Health Sciences, Campbell University, Buies Creek, NC 27506, USA; easimonds@gmail.com (E.A.S.); kpgobenciong1119@email.campbell.edu (K.A.P.G.); jiroutekm@campbell.edu (M.R.J.); 2OpSAFE International, Tokyo 198-0013, Japan; jwilson@opsafeintl.com; 3The Warren Alpert Medical School, Brown University, Providence, RI 02903, USA; nicole_nugent@brown.edu; 4Joan C. Edwards School of Medicine, Marshall University, Huntington, WV 25701, USA; 5Department of Medicine, University of North Carolina, Chapel Hill, NC 27599, USA; 6School of Social Work, University of Washington, Seattle, WA 98105, USA

**Keywords:** mental health, mental health first aid, pediatric mental health, pediatric trauma, psychological first aid

## Abstract

Background: There is worldwide consensus that providing secondary prevention to promote resilience and prevent mental health concerns after a disaster is important. However, data supporting this kind of intervention is largely lacking. The current study evaluates the effectiveness of OperationSAFE, an early intervention for children after community-wide trauma. Methods: Secondary data analyses of data collected during 158 OperationSAFE camps (a five day camp with a curriculum focused on coping with stressors) in five countries and ten disasters between 2015 and 2020 were performed. Data on child trauma-related functioning/well-being were collected by an OperationSAFE in-house developed symptom checklist and completed by counselors about children on the first and last day of the 5-day camp. Results: A total of 16,768 children participated in the camps (mean age 9.4 ± 2.36; 50% male). Trauma-related functioning/well-being improved from day 1 to day 5 (b = 8.44 ± 0.04; *p* < 0.0001). Older children improved more (b = 0.22 ± 0.01; *p* < 0.0001). Children in man-made ongoing trauma (war/refugees) situations responded stronger than those after natural disasters (b = 2.24 ± 0.05; *p* < 0.0001). Negligible effects for gender and the number of days between a traumatic event and the start of camp were found. Conclusions: This is the first study to show in a large and diverse sample that secondary prevention to promote resilience and prevent mental health concerns after a disaster for children is associated with improvements in trauma-related functioning/well-being. Delaying delivery of the intervention did not affect outcomes. Given the uncontrolled nature of the study and lack of long-term outcomes, more studies are needed to corroborate the current findings.

## 1. Background and Introduction

Worldwide it is estimated that in the last decade approximately 1.5 billion people have been impacted by natural disasters [1]. The impact of disasters on communities is substantial, particularly for children whose primary support system—parents and other adults—needs to focus initially on securing necessities, such as housing, food, and water, and later on rebuilding. The impact of childhood trauma can persist into adulthood, adversely impacting mental health and quality of life for years to come [1,2]. Although most initial post-traumatic stress symptoms resolve in the weeks after the trauma, approximately 10% of children develop chronic severe symptoms of post-traumatic stress disorder, with a third showing moderate symptoms that persist even a decade later [3]. Importantly, disaster exposed children and adolescents are at increased risk of other negative outcomes of trauma such as depression and anxiety [4,5]. Relatively few studies have described efforts to prevent post-traumatic stress disorder and other mental health consequences in trauma-exposed children and adolescents, and even fewer have evaluated these efforts in disaster settings [5,6,7,8]. To prevent adverse sequelae of trauma and promote early adaptation after a traumatic event, the World Health Organization and many other government and non-profit organizations recommend implementing help to ameliorate the immediate psychological reactions after a disaster [8,9].

“First and second level” or secondary preventions—interventions implemented in the initial days to weeks post-trauma, such as Psychological First Aid—have been developed to provide broad support to trauma-exposed adults and children during or immediately after a traumatic event, irrespective of symptoms [4,10,11,12,13]. Existing psychological secondary preventions have been developed to address the immediate psychological distress, enhancing existing support and addressing acute concerns, but are not intended to involve the provision of ongoing mental health treatment. The rationale for this approach is partly based on the idea that a majority of disaster-exposed individuals will not go on to develop post-traumatic stress disorder and other negative outcomes of trauma, and therefore the goals of early interventions are to provide short-term support and refer for further treatment as needed. Early secondary preventions such as Psychological First Aid are widely adopted: over twenty countries now consider secondary preventions as an essential part of disaster response and deliver formal training to clinicians, laypeople, and first responders [14].

Despite the widespread implementation of preventative interventions, the evidence for effectiveness is scant [5,15,16,17]. Research has shown that training improves the knowledge and recognition of mental health problems in those who provide the support, but data on recipients is lacking [18]. A few studies have been completed outside of immediate disaster relief. For example, in a small non-controlled trial among twenty children undergoing stressful life experiences, Psychological First Aid was implemented by a school nurse [19]. Over an 8-week period, improvements in depression, post-traumatic stress symptoms, social support, and school connectedness were observed. Building on this work, Morgan and colleagues [20] randomized 384 parents to receive Mental Health First Aid or Red Cross Provide First Aid training to help their adolescents if future traumatic experiences occurred. Compared to the Red Cross training, parents in the Mental Health First Aid group reported increased knowledge about mental health problems and how to respond should they occur. However, no differences between the groups were found in quality of parental support or the proportion of cases of adolescents with a mental health problem over a 1–2-year follow-up period. Thus, the study found no evidence that providing parents knowledge of mental health issues and how to respond to them improves adolescents’ mental health over time. However, not all of these adolescents can be expected to have endured a traumatic experience during the 2-year follow-up.

In adults, the evidence is similarly scant. A randomized controlled trial of Psychological First Aid among adult crime victims found no evidence of improved trauma symptoms or psychological distress [21]. Women saw accelerated improvement in psychological distress and hence they felt better sooner with Psychological First Aid. A small randomized controlled trial among 40 young adults found a 30-minute post-intervention Psychological First Aid decrease in anxiety [22] but did not assess longer term symptoms. Thus, few large, well-designed studies exist testing Psychological First Aid in either children or adults. None of these studies tested the effectiveness of interventions after a community exposure to a traumatic event.

The current work is the first to examine changes in trauma symptoms with an early intervention—OperationSAFE camps (details given in methods)—for children after community-wide trauma. Ethically, OperationSAFE could not exclude children from help; therefore, the study employs a pre-post evaluation of OperationSAFE camps. We tested the implementation of OperationSAFE in a large diversity of settings, including five different countries, ten different traumatic events, and both man-made (e.g., wars, refugees) and natural disasters (e.g., earthquakes, hurricanes, and tsunamis).

## 2. Methods

This study reports analyses of data collected for program evaluation of OperationSAFE (http://opsafeintl.com/ (accessed on 18 January 2019)). The mission of OperationSAFE is to aid in the care and mental health of children who have experienced trauma [1]. OperationSAFE provides early post-disaster intervention, implemented through a camp setting, for children in partnership with local programs. They leverage the concept of vacation bible schools, which are commonplace in faith-based ministries around the world. This format makes the deployment of a camp relatively straightforward, given that the model is prevalent worldwide.

For this study, we used existing program evaluation data from OperationSAFE. Camps were deployed after 10 disasters in 5 countries. In one country, camps were underway for one typhoon when the area was hit by another typhoon. Given the closeness in timing and location, we could not separate the camps specific to each typhoon. Some existing camps were expanded to accommodate those affected while new ones were created for previously unaffected areas. Some of these children, in fact, faced a natural disaster twice. Therefore, we combined these data into one event. A total of 158 campsites were run across these 9 disasters, between the years 2015 and 2020.

### 2.1. OperationSAFE Content

#### 2.1.1. OperationSAFE Aim and Overview

OperationSAFE is a psychosocial intervention designed to reduce post-traumatic stress symptoms in children exposed to community trauma aged 6–12 years old. It also focuses on strengthening their connections to support in their community. OperationSAFE developed a protocol that consists of five pillars: promoting safety, calm, a sense of self-efficacy, connectedness, and instilling hope. These pillars are consistent with core components essential to emotional recovery for disaster-exposed children as described in recent guidelines [23]. The curriculum, which is deployed in all camps, is designed to be engaging and child friendly and includes camp and counselor management, identification and understanding of post-traumatic stress symptoms (including post-traumatic stress disorder, depression, and anxiety), assessment and reporting of symptoms, the basics of preventative early intervention in children, skills for self-care, and the OperationSAFE stories and content. The OperationSAFE characters who tell the stories are from the Antarctic, and “Pete the Penguin” is the protagonist (see Figure 1); his continuity throughout the camp experience has been used to facilitate coping with feelings. This approach is ideally suited to both those communities with high literacy rates as well as those with oral traditions. The activities taught at the camp include stories, games, arts and crafts, and songs. The camps allow participants who faced a disaster to make new friends, overcome their fears by sharing experiences, and regain hope. Children attend camps for at least 4 h each day, providing a safe space and supervision while parents deal with the disaster’s aftermath. OperationSAFE materials are developed for children aged 6–12 years old, but children outside of this age range were not turned away for humanitarian reasons.

#### 2.1.2. OperationSAFE Training and Cultural Adaptation

Local community leaders and volunteers deliver the intervention to increase cultural adaptability. These locals are often not trained in mental health aid and therefore receive 3-day training consisting of OperationSAFE curriculum training (including a manual), information on trauma, its effects on children, and how they as a community can support the children through it. Currently, the OperationSAFE program is translated and culturally adapted into Sichuanese, Tibetan, Japanese, Korean, Haitian Creole, Tagalog, Waray, Cebuano, English, and Ukrainian [1]. Cultural adaptation of the curriculum is important because the local community is an important part of a child’s resilience. Connections begun during the camp can continue as the children recognize them in the community, at school, or in their religious organizations, providing the children with ongoing sources of support.

#### 2.1.3. OperationSAFE Format and Content

The intervention format is copied after bible summer school, which is familiar in many communities and provides access to resources such as venues and volunteers. This allows children a safe space while their parents/caregivers deal with the many tasks in the aftermath of a community disaster. Despite the previously described association between religion and resilience [24], the intervention itself does not include a religious component, and children of all religions/backgrounds are welcomed.

When the children arrive at the camp venue, they are organized into small groups of five children with one volunteer leader. They will stay with this small group throughout the week for each activity. For children who have been evacuated from their neighborhoods, this allows them to form a friend group. For all children, the shared activities, discussions, and experiences enable them to connect more deeply with peers and understand that they had some of the same emotions and reactions as their peers. The peer group is an essential part of resilience for children.

Once the program begins, the children spend three hours a day with their peers, the volunteer leader, and the larger group of children and leaders in the camp. Over the week, the camp emphasizes five themes with the children: safety, calming, hope, self and community-efficacy, and connection. These are expressed in a way that makes sense to children. For safety, the theme “I am not alone” (see Figure 1) is used as a core element of safety for a child demonstrating that someone is there to protect them. This is taken deeper as the children find out that they are not alone in the emotions they felt during the traumatic event and after. For calming, the theme “Everyone is Important” (see Figure 1) is used to emphasize listening to everyone. Children often feel that their story is unimportant compared to adults’ concerns and become frustrated because they are not understood. For hope, the theme “Follow and Believe” (see Figure 1) is used with an emphasis on trusting leaders and restoring normalcy. Getting back to normal activities, such as attending school, doing chores, playing, and studying, helps restore hope that things will get better. For self and community-efficacy, the theme “Be Strong and Courageous” (see Figure 1) is used with an emphasis on being brave enough to ask for help. Children are not simply victims or helpless and become active participants in encouraging one another amid the crisis. For connectedness, the theme “You are Loved” (see Figure 1) is used to emphasize friendship and inclusion. As children struggle with their feelings after exposure to traumatic events, they can find relationships difficult, including those who are struggling as a community reminds all of the children that they are accepted as well.

Each day of the program begins with an opening event held as a large group with all of the children attending in their own small group. During this opening event, the theme of the day is introduced, a new song is taught, and the themes and songs of previous days are repeated. Each song has motions, and the children are encouraged to sing along and dance by their leaders from the front. For many of the children, adults around them have been serious, fearful, and anxious. From the opening event, children see from the actions and expressions of the adults that they can play, laugh, and enjoy themselves, i.e., they are safe. Songs also can have a calming effect, allowing children to sense rhythm, melody, and repetition. At the end of the day, there is a similar closing event that brings all the children together once again.

The rest of the day is broken into five activity stations for the children to attend with their small group. Each activity is designed to reinforce the theme of the day. However, the activities themselves also help the children recover by using all of their senses and emotions. The order of the stations changes depending on which group the children are in; however, each group will participate in all five by the end of the day:(1)The story station tells the adventure of a penguin who is separated from his family and all of the emotions that he goes through (see Figure 1). It introduces the theme of the day and ways to cope. It also provides an opportunity to be able to talk through difficult things with their counselor/peer group.(2)The craft station allows children to use their hands and their creativity to express themselves related to the theme of the day and/or their response to trauma. The focus is on building self-efficacy (having pride in being able to make something themselves) and emphasizing finding support (when things are too difficult).(3)The snack station is an opportunity to bond with peers.(4)The game station reinforces the day’s theme with kinesthetic play. Some children are much more likely to remember something if they have a chance to act it out practically.(5)The mindfulness station teaches and practices simple calming exercises that they can repeat on their own even after the intervention is finished. These include diaphragmatic breathing, stillness, and awareness of their own bodies to help them regulate their emotions.

This mix of activities allows children to use all of their senses considering that a child’s response to traumatic experiences is not primarily cognitive, but also expressed through behavior, emotions, reactions in their body, and relationships with others.

### 2.2. Measures

All measures used for OperationSAFE camps can be found here: https://ee.kobotoolbox.org/x/YvM7 (accessed on 18 January 2019). Measures were completed by camp counselors (Appendix A). The primary outcome measure for OperationSAFE is post-trauma functioning and well-being.

### 2.3. Trauma-Related Functioning and Well-Being

OperationSAFE developed a 7-item measure designed to align with indicators of post-trauma functioning and wellbeing, especially those related to changes in mood, arousal, and activity. These include scores of each child’s: general health status, attention level, activity level, affect, sociability, general appetite, and reaction to conflict/distress. All items were scored on a 5-point Likert scale, with lower scores indicating worse functioning/well-being post-trauma. For example, general health was scored from 1 = poor to 5 = excellent. Total scores were calculated for day 1 and day 5 by summing all items measured at each time point (range of scores: 7–35). All items were translated into local languages. Children’s ratings were completed by the counselor at the start (day 1 of camp) and end of camp (day 5 of camp). This tool was developed in part with the KoBo Toolbox, (KoBo Inc., Cambridge, MA, USA) from the Harvard Humanitarian Initiative (www.kobotoolbox.org/ (accessed on 18 January 2019)), an open-source suite of tools for data collection and analyses in humanitarian emergencies. Given the measure was developed for quality improvement purposes, it was not fully validated; however, it contains at least face validity based on the content of the items.

### 2.4. Data Analyses

SAS statistical software (SAS Institute, Inc., Cary, NC, USA) was utilized to fit a series of marginal models with generalized estimating equations (PROC GENMOD SAS Institute, Inc., Cary, NC, USA) to assess the predictive value of OperationSAFE (i.e., the effectiveness of counselor) on total trauma functional and well-being score (0 to 35 scale). As post-trauma functional and well-being scores were available for each child both at the start and end of camp, timepoint was nested within child which was nested within counselor.

Marginal models are a type of multilevel model that account for a nested multilevel data structure with correlated observations at each level [25]. The first model constructed contained the timepoint predictor variable, controlling for disaster location as a fixed effect (five disaster locations: Cotabato, Visayas, Mindanao, Mongolia, and Nepal) using the available data from all locations. This model was then rerun with the addition of the control variables child age, gender, and disaster type (natural vs. man-made). Camp delay (the duration in days between a disaster occurrence and the start of a camp) was thought to be important, but camp delay information was only available for natural disasters. In man-made disasters such as wars and refugee camps, trauma was often ongoing so camp delay could not be measured. Thus, the initial model with the timepoint predictor variable, controlling for disaster location as a fixed effect, was rerun including camp delay (in days) only on that subset of the data.

The correlation structure for each model was assessed with quasi-likelihood under the null (QIC) criteria and the compound symmetry correlation was selected. Model parameter estimates (±standard error) along with 95% confidence intervals and associated *p*-values are reported. The parameter estimates are slope estimates, describing the linear relationship between a predictor variable and total trauma functioning, and well-being score. Positive slope estimates suggest children’s total post-trauma score increased (psychological well-being improved), on average, by the number shown (on a 35-point scale) as each predictor variable increased by one unit. While the authors have included *p*-values for all inferential results, following the recent recommendations of the American Statistical Association, no cutoff for a *p*-value is indicated as the basis for a decision about the meaningfulness/importance of an effect [26]. Further, the large study size (*n* = 16,768) is likely a factor in (small) *p*-values, corresponding to results that may not be clinically meaningful.

Given that data was collected online, no questions could be skipped, hence there is no missing data in the dataset.

### 2.5. Human Ethics

As the data was collected for program evaluation, no human subject ethics requirement was obtained at that time. The Campbell University IRB approved the use of the data for secondary data analysis (IRB# 587 obtained on 4 September 2020).

## 3. Results

### 3.1. Demographics

A total of 158 campsites were run for ten disasters in five countries between the years of 2015 and 2020. A total of 16,768 children participated in the camps. OperationSAFE materials are developed for children aged 6–12 years old, but children outside of this age range were not turned away for humanitarian reasons. The vast majority of children were within this age range (14,914; 88.9%). Table 1 provides a demographic breakdown of the sample. Half of the sample was male, and half of the sample experienced a natural disaster. There were no differences in age (M = 9.41 ± 2.40 vs. M = 9.34 ± 2.33; *p* = 0.054) or gender (male = 4101 (49.72%) vs. male = 4278 (50.22%); *p* = 0.516) for those who were exposed to natural vs. man-made disasters, respectively. No information on the race of the children was collected, but all countries were within Asia.

### 3.2. OperationSAFE Program Effectiveness

All of the combined disaster data finds that post-trauma functioning/well-being sum score improves from day 1 to day 5 after controlling for disaster location (b = 8.44 ± 0.04; *p* ≤ 0.0001; see Figure 2). Appendix B shows the change from day 1 to day 5 scores for each separate trauma item (general health status, attention level, activity level, affect, sociability, general appetite, and reaction to conflict and distress) across all subjects from all sites. All these items showed a similar improvement over the 5-day period.

In one camp location, Mindanao, the response to camp was overwhelming with 400 more children showing up than expected. This may have overburdened the counselors and affected how well each counselor was able to evaluate functioning/well-being in campers. Therefore, we compared the four largest camp locations, including Mindanao, separately. As can be seen in Table 2, similar improvements were found across all four camp locations.

The effect of camp participation on well-being/functioning over time increased slightly with age of child (b = 0.22 ± 0.01; *p* < 0.0001), while the effect for gender was negligible (b = −0.07 ± 0.06; *p* = 0.1992). Changes in post-traumatic functioning/well-being observed over participation in OperationSAFE were greater for man-made disasters than natural disasters (b = 2.24 ± 0.05; *p* < 0.0001). Among those experiencing natural disasters, the effect of camp delay was essentially non-existent (b = 0.004 ± 0.0003; *p* <0.0001). This variable was not tested in man-made disasters, such as wars and refugee camps, given that these are often ongoing during OperationSAFE camp experiences. Therefore, the camp delay could not be measured.

## 4. Discussion

This is the first study to systematically collect data related to secondary prevention programming in a large sample of children across various disasters such as wars, refugee camps, earthquakes, typhoons, and tsunamis. OperationSAFE camp participation for children resulted in an improvement in trauma-related functioning/well-being scores with large effect sizes. This change in symptoms was similar for boys and girls and slightly greater with increased age. The OperationSAFE camps focus on re-evaluating stressors and encouraging cognitive coping strategies to deal with largely unchangeable circumstances (at least from the child’s point of view). From a developmental perspective, coping becomes more diversified with age. Younger children (ages 5–7) rely primarily on behavioral strategies such as avoidance; children in middle childhood increasingly rely on cognitive strategies [27], which may explain why older children respond better to OperationSAFE camp participation. Future camps should consider improving content for children aged 5–7 to increase efficacy in younger ages.

Although children, in general, showed improvements with OperationSAFE, camp participation was especially beneficial to children who had experienced man-made disasters such as wars, which are known to be associated with the highest risk of post-traumatic symptomatology. The time between a disaster and the start of a camp had a non-meaningful effect on trauma functioning/well-being scores. This is encouraging news as some camps were conducted >6 months post-disaster. Demonstrating the usefulness of a mental health aid program despite an increase in camp delay indicates that interventions can help children potentially many months after a traumatic event. These findings are encouraging and suggest that secondary intervention can be helpful for children across multiple settings and circumstances.

Early intervention after a traumatic event is important as approximately 40% of children are at risk of long-term post-traumatic stress symptoms [3]. Early interventions such as Psychological First Aid have broad support from governments and experts, but data on the efficacy/effectiveness of post-disaster interventions for children is largely lacking. Previous studies with Mental Health First Aid have shown that training of volunteers who are not clinicians increases their mental health knowledge and reduces stigma [18], but these studies did not show that this knowledge translated into better adjustment for children. A few studies examining early interventions, such as Psychological First Aid and Mental Health First Aid for individual trauma victims (not community-level disasters), raised contradictory results regarding the effect of these programs on mental health [19,20,21]. The current study is the first to add evidence that early intervention is associated with improved functioning and well-being in children after disasters affecting large communities.

The current study only evaluated the immediate effect of the OperationSAFE program. It is important to extend future studies to include long-term data. This is particularly important because programs focused on resilience after trauma are ultimately designed to prevent future negative mental health outcomes. Therefore, long-term data is needed. After a disaster, it is expected to see functioning and well-being scores improve over time (also known as regression to the mean). In a study among Thai children exposed to a Tsunami, the prevalence of PTSD was high 6 weeks after the disaster (57.3%) which declined to 7.6% two years post-disaster [28]. These children received integrated welfare, including mental health help if needed, over this time period which likely also had a beneficial effect. However, these numbers suggest that post-traumatic growth and resilience are likely the rule rather than the exception. To account for this effect, not only long-term data collection but also the incorporation of a control group is of utmost importance. The current study is a secondary data analysis of data collected for project evaluation purposes and therefore not designed to reduce the risk of bias in study design. Due to ethical considerations regarding withholding intervention from children in need, OperationSAFE did not turn away any children, and hence no control group was included. Future studies should be planned with answering the question of efficacy in mind.

The current study is not able to test which parts of the OperationSAFE program are most likely to yield positive benefits. However, resilience and post-traumatic growth are considered to be multifactorial, and any programming should address trauma from multiple levels (personal, family, and community-wide) [29]. OperationSAFE focused primarily on enhancing individual factors to increase resilience. By engaging counselors from the community, it indirectly addressed community factors by increased connection to people in the community who can offer a child help after the camp has ended. It also ensured that adult community members were trained in trauma awareness, which helps responses to the aftermath of a disaster. OperationSAFE did not include family-level interventions, even though the family can be argued to be one of the main factors enhancing child resiliency [29]. However, in the aftermath of a disaster family members are often busy securing first-order needs, such as shelter, food, clothes, etc., as well as dealing with their own reactions to the trauma. It would be ideal if all families could rely on organizations that provide these first-order needs as well as offer psychological first aid to all adults. A child’s resilience will be increased by helping their caregivers. Thus, child psychological first aid should ideally be planned as part of an overall community response to a disaster.

The study had several other limitations. First, child functioning/well-being was evaluated by counselors, not the children themselves. This proxy report can introduce bias. Furthermore, the measure used was developed for program evaluation and was not validated before use. Future studies should validate this measure against other measures of child functioning, well-being, and trauma. For example, counselors may wish to believe that children improve in a program they delivered. Second, no data is available on the well-being of children before the disaster. This is of course almost impossible since we are unable to predict when disasters hit. Right before an impact, anticipatory stress and preparatory behaviors (e.g., leaving home to seek shelter) already induce distress. We acknowledge that many of the children in the current study lived in impoverished and underserved areas before the disaster, where exposure to adverse childhood experiences is expected. However, it is expected that a community-wide disaster would only increase distress in children already exposed to adverse experiences, arguing the need for intervention, especially in such a group. Third, despite the implementation of OperationSAFE in multiple countries, these were all located on the Asian continent, potentially limiting generalizability to other locations. Future studies are needed that compare OperationSAFE and other secondary post-disaster interventions to a control condition, collect long-term data from children, and include non-Asian countries.

In sum, the current study demonstrates that OperationSAFE camps, an early preventative post-disaster intervention, are associated with better functioning and well-being scores in children exposed to a community disaster. Given that the intervention was tested in over 16,000 children in five countries and 10 disasters, this provides reassurance that the findings are generalizable to various disasters and settings. Children should receive equal access to mental health aid in addition to physical first aid post-disaster.

## Figures and Tables

**Figure 1 children-09-00951-f001:**
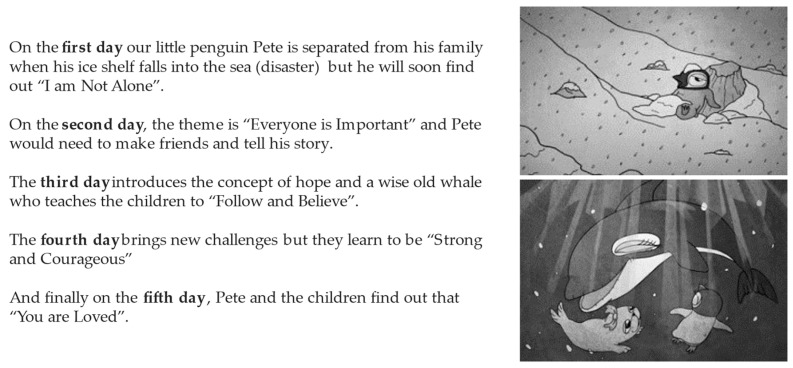
OperationSAFE’s Pete the Penguin teaches 5 principles of MHFA.

**Figure 2 children-09-00951-f002:**
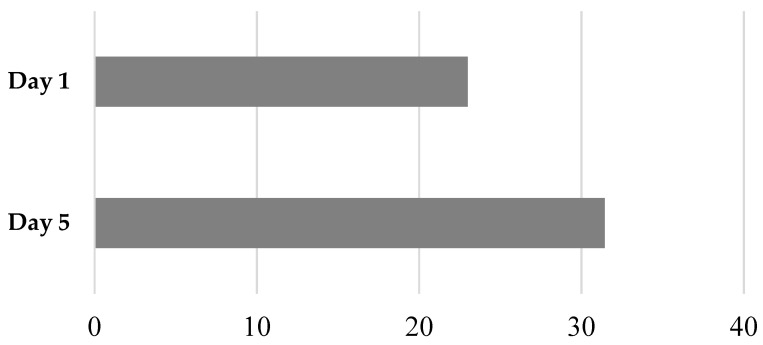
Improvement in trauma symptoms with MHFA.

**Table 1 children-09-00951-t001:** Sample characteristics.

	Mean (±S.D.)
	or
	Frequency (%)
Age	9.4 (± 2.36)
Gender: male	8379 (50)
Number of children per counselor (*n* = 566 counselors)	8 (± 20)
Number of children per campsite (158 total sites)	107 (± 36.42)
	Range = 2–642
Children affected by each disaster type	
Man-made (*n* = 2 disasters)	8519 (51)
Natural (*n* = 7 disasters)	8249 (49)
Number of days between disaster and start of camp	260.6 ± 137.81

**Table 2 children-09-00951-t002:** Improvement in trauma symptoms by location for the four largest disasters/campsites.

Campsite ^a^	Type of Disaster	Β ^b^ (±SE)	*p*-Value
Visayas (*n* = 2508)	Typhoon	−1.53 ± 0.19	<0.0001
Mindanao (*n* = 13,092)	Conflict	−0.73 ± 0.16	<0.0001
Mongolia (*n* = 222)	Poverty	2.46 ± 0.27	<0.0001
Nepal (*n* = 915)	Earthquake		

^a^ Parameter estimate for each location’s camps relative to Nepal location camps. ^b^ Estimated average increase (decrease) in day 5 score relative to day 1 score.

## Data Availability

The data is not publicly available. OpSAFE will share de-identified individual data with investigators on a case-by-case basis. Interested investigators should contact OpSAFE with a methodologically sound proposal that addresses a novel question. Investigators will be required to sign an agreement/contract before gaining access to the data.

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
