# Peer review of "Trauma Functioning and Well-Being in Children Who Receive Mental Health Aid after Natural Disaster or War"

_children, 2022, doi:10.3390/children9070951_

Round 1

Reviewer 1 Report

The reviewer has read with pleasure the content of the manuscript, which addresses one of the most important research factors regarding “Trauma functioning and well-being in children who receive mental health aid after natural disaster or war.”

 In the opinion of the reviewer, the text is valuable and could be reconsidered after major revision. Therefore, it is recommended improve  of text or/and detailed clarification response to reviewer's raised issues:

- the title of the manuscript is not fully adequate to the findings of research in the war issue;

- line 25-26 and 202-204 (quote) “ Trauma-related functioning/well-being improved from day 1 to day 5 (b=8.44±0.04; p< 0.0001)”. In fact, such result is rare in scientific publications. Therefore, all measures used and confirming such results are very appreciated as supplementary materials to this manuscript. Please provide all of them;

- line 102-104; phrase (quote) “In one country, camps were underway for one typhoon when the area was hit by another typhoon. Given the closeness in timing and location, we combined these data into one event.” Please explain why the authors did not use these data to evaluate each typhoon event separately to determine how deeply each typhoon had a negative effect on mental health. Combine of such  results could give a new sign on separate typhoon and cumulative typhoons mental health outcomes as natural disasters. Unfortunately, the authors did not focus on this issue;

- All measures used for OperationSAFE camps must be included as Supplementary Materials to this manuscript;

- line 18-190: It is missing alphabetical/numerical approval code applied to  The Campbell University IRB;

- missing demographic data related to natural disaster , and to experienced man-made disasters such as wars;

- missing strengths and limitations paragraph;

- missing conclusion paragraph;

- missing Institutional Review Board Statement;

- reference numbers are incorrectly inserted in the manuscript;

- references must be re-edited to comply with the journal guideline.

Author Response

Please see attached document entitled Reviewer 1

Reviewer 2 Report

This is an important and timely paper on the effectiveness of early intervention programs for supporting children after community-wide trauma events, using the international program OperationSAFE as a case study.

In general, this paper uses interesting data and intends to offer a kind of non-controlled natural case study of how intervention programs might help, and to this extent the paper has merit in providing some general orientation in how and what to measure in terms of the efficacy of these programs. Thus, the paper offers a useful initial pilot-like study for future surveys on this kind program. However, the paper does read a bit like a somewhat undigested reflection on these 5 camps, but without much detail on what actually was happening in the camps.

Moreover, one of the major limitations--as the authors also mention--is that we only have the evaluations from the staff, and not from the children themselves. While this is impossible to rectify retroactively, it does implore future research to find some way to include the children's perspectives; and indeed, reflection on time in such a camp could be integrated into the therapeutic process itself in the camp programs. Perhaps the authors could also extend their Discussion to include some suggestions for how to increase the effectiveness of the camps as well as future research.

It would also be helpful for the authors to address the potential influence of religious beliefs as a source of resilience among these different sites, as strength of religious belief has been seen as correlated in some ways to resilience and perspectives on disasters. Particularly in The Philippines, where religious belief is strong, survivors' response to disasters is related to how they make sense of it through the lens of theodicy. In contrast, in Japan, religious beliefs are not as strongly held by most of the population, and so the "meaning" of disasters and the ways that survivors make sense of disasters is different. This would likely have some correlation with post-disaster resilience and mental health outcomes.

Below are some more specific comments:

After the Background section, it might be helpful to explain in more detail the specific activities that were conducted each day at the Operation SAFE camps. I understand that there was local variation, but it would be helpful to provide an explanation (or perhaps a Figure) outlining the schedule for an "average" camp program; this could be the information from Appendix 2, or a synthesized version.

Minor Points:

Figure 1 has a mistype: "...a wise old whale who teaches [them] children" --> "who teaches the children"

Language editing is needed throughout to smooth the grammar and catch typos.

The naming of the appendices in the text and in the actual Appendices is inconsistent (sometimes Appendix A, sometimes Appendix 1, and then there is an Appendix 2).

Also, where do the scores come from in Appendix 1/A? Is the generalized across all of the case study sites, is it from a single site, or is this the total program's international data?

Author Response

Please see attached File Reviewer 2 

Round 2

Reviewer 1 Report

Despite the author's revised manuscript, not all of the reviewer's recommendations were taken into account or omitted. For example:

-       missing “conclusion” paragraph.

Readers must clearly see paragraph titled “Conclusion.” In present version of manuscript is so difficult to find phrases related to conclusion

- missing “strengths and limitations” paragraph.

Readers must clearly see paragraph titled “strengths and limitations.” In present version of manuscript is so difficult to find phrases related to strengths and limitations.

In reference to difficulties to establish both “conclusion” paragraph and “strengths and limitations” paragraph , reviewer recommended  to see how both of the paragraphs are established i.e. Bassi, G.; Mancinelli, E.; Boldrini, B.; Mondini, G.; Ferruzza, E.; Di Riso, D.; Salcuni, S. Perception of Changing Habits among Italian Children and Adolescents during COVID-19 Quarantine: An Epidemiological Study. Children 2022, 9, 806. https://doi.org/10.3390/children9060806

- reference numbers are incorrectly inserted into main text of the manuscript;

In reference to difficulties about incorrectly inserted reference numbers into main text of the manuscript , reviewer recommended  to see how it must be done correctly i.e. Nuñez-Fadda, S.M.; Castro-Castañeda, R.; Vargas-Jiménez, E.; Musitu-Ochoa, G.; Callejas-Jerónimo, J.E. Impact of Bullying—Victimization and Gender over Psychological Distress, Suicidal Ideation, and Family Functioning of Mexican Adolescents. Children 2022, 9, 747. https://doi.org/10.3390/children9050747

Additionally, .Authors’ quoted response to reviewer’s recommendation round # 1 must be included in paragraph “ strengths and limitations”: 

“ We agree that each disaster by itself could give interesting results. However, due to the proximity of the typhoons and ongoing camp activity at that time, we could not separate the camps specific to each typhoon as some existing camps were simply expanded to accommodate those affected while new ones were created for previously unaffected areas. Some of these children, in fact, faced a natural disaster twice. This made separating camps and children within camps for these two events impossible. Therefore, we felt that combining the data from these two events was the only valid solution of those available to us. Please do note that our main analyses combine all data across all sites, while statistically controlling for site, the most granular level we could attain with the existing data.”

Author Response

We thank the reviewer for their additional comments on our paper.  Responses are outlined below in italics.

Despite the author's revised manuscript, not all of the reviewer's recommendations were taken into account or omitted.

We took effort to respond to each of the reviewer’s concerns. In cases, where we did not make changes based on the reviewer’s feedback, we explained why we chose to do so. We have expanded our explanation below.

For example:

-       missing “conclusion” paragraph.

Readers must clearly see paragraph titled “Conclusion.” In present version of manuscript is so difficult to find phrases related to conclusion

As explained in the previous reply, we are not missing a conclusion paragraph. The last paragraph is clearly a conclusion paragraph. Our last paragraph even starts with the words ‘In sum’.

The journals guidelines include the following: “Conclusions: This section is not mandatory but can be added to the manuscript if the discussion is unusually long or complex.”

  As we are not in omission of the journal guidelines nor any of general scholarly writing expectations, and we include a conclusion paragraph – we have made no additional changes based on this comment.

- missing “strengths and limitations” paragraph.

Readers must clearly see paragraph titled “strengths and limitations.” In present version of manuscript is so difficult to find phrases related to strengths and limitations.

As explained in our previous reply to the reviewers, we have included three paragraphs of limitations. One of the paragraphs even starts with the sentence “The study has several other limitations”. Combining these three in one paragraph would make the discussion more difficult to read. To preserve readability of the text, we respectfully decline making the changes the reviewer suggests.

In reference to difficulties to establish both “conclusion” paragraph and “strengths and limitations” paragraph , reviewer recommended  to see how both of the paragraphs are established i.e. Bassi, G.; Mancinelli, E.; Boldrini, B.; Mondini, G.; Ferruzza, E.; Di Riso, D.; Salcuni, S. Perception of Changing Habits among Italian Children and Adolescents during COVID-19 Quarantine: An Epidemiological Study. Children 2022, 9, 806. https://doi.org/10.3390/children9060806

We appreciated the reviewers’ suggestion of a system they personally like for their manuscripts. We are under no obligation to adhere to a system the reviewer prefers – but is not required by the journal. Therefore, we have made no additional changes.

- reference numbers are incorrectly inserted into main text of the manuscript;

In reference to difficulties about incorrectly inserted reference numbers into main text of the manuscript , reviewer recommended  to see how it must be done correctly i.e. Nuñez-Fadda, S.M.; Castro-Castañeda, R.; Vargas-Jiménez, E.; Musitu-Ochoa, G.; Callejas-Jerónimo, J.E. Impact of Bullying—Victimization and Gender over Psychological Distress, Suicidal Ideation, and Family Functioning of Mexican Adolescents. Children 2022, 9, 747. https://doi.org/10.3390/children9050747

We made changes based on this feedback in our first round when some references were in brackets and others not. We check the numbering then and now again, which is correct. We moved all references numbers before the period instead of after the period, in case that is what the reviewer is referring to. If anything else is wrong, according to the reviewer, please be more specific.

Additionally, .Authors’ quoted response to reviewer’s recommendation round # 1 must be included in paragraph “ strengths and limitations”:  

“ We agree that each disaster by itself could give interesting results. However, due to the proximity of the typhoons and ongoing camp activity at that time, we could not separate the camps specific to each typhoon as some existing camps were simply expanded to accommodate those affected while new ones were created for previously unaffected areas. Some of these children, in fact, faced a natural disaster twice. This made separating camps and children within camps for these two events impossible. Therefore, we felt that combining the data from these two events was the only valid solution of those available to us. Please do note that our main analyses combine all data across all sites, while statistically controlling for site, the most granular level we could attain with the existing data.”

We made the following changes to the methods section. “In one country, camps were underway for one typhoon when the area was hit by another typhoon. Given the closeness in timing and location, we combined these data into one event.” We changed this to: “In one country, camps were underway for one typhoon when the area was hit by another typhoon. Given the closeness in timing and location, we could not separate the camps specific to each typhoon.  Some existing camps were expanded to accommodate those affected while new ones were created for previously unaffected areas. Some of these children, in fact, faced a natural disaster twice. Therefore, we combined these data into one event.”